# Characterizations of Newly Isolated *Erwinia amylovora* Loessnervirus-like Bacteriophages from Hungary

**DOI:** 10.3390/v17050677

**Published:** 2025-05-06

**Authors:** Elene Lomadze, György Schneider, Szilvia Papp, Dominika Bali, Roberta Princz-Tóth, Tamás Kovács

**Affiliations:** 1Enviroinvest Corp., Kertvaros St. 2, H-7632 Pecs, Hungary; elene.lomadze@enviroinvest.hu (E.L.); papp.szilvia@enviroinvest.hu (S.P.); 2Department of Medical Microbiology and Immunology, Medical School, University of Pécs, Szigeti Str. 12, H-7624 Pecs, Hungary; schneider.gyorgy@pte.hu; 3Biopesticide Ltd., Kertvaros St. 2, H-7632 Pecs, Hungary; d.bali@biopesticide.eu (D.B.); r.toth@biopesticide.eu (R.P.-T.)

**Keywords:** fire blight, *Erwinia amylovora*, bacteriophage-based biocontrol, environmental stability

## Abstract

This study explores alternative methods to combat bacterial infections like fire blight caused by *Erwinia amylovora* (Ea) using bacteriophages as potential antimicrobial agents. Two lytic phages, Ea PF 7 and Ea PF 9, were isolated from apple samples and classified as Loessnervirus-like based on their genomes. Both phages showed strong efficacy, lysing 95% of the tested 37 Ea strains. They inhibited bacterial growth for up to 10 h, even at low infection rates. The phages had a short latent period of 10 min and produced high burst sizes of 108 and 125 phage particles per infected cell. Stability tests revealed that both phages were stable at moderate temperatures (37–45 °C) and within a pH range of 4–10. However, their viability decreased at higher temperatures and extreme pH levels. Both phages exhibited notable desiccation tolerance and moderate resistance to UV-B radiation during UV testing. The phages were exposed to carefully controlled irradiation, considering factors like lamp type, radiation intensity, exposure time, and object distance. This method introduces a complex approach to research, ensuring repeatable and comparable results. These findings suggest that Ea PF 7 and Ea PF 9 hold promise as antimicrobial agents for therapeutic and biotechnological applications, potentially helping to combat antibiotic resistance in the future.

## 1. Introduction

*Erwinia amylovora* (Ea), is a Gram-negative bacterium causing a devastative disease, so-called fire blight, in various plants, such as apples, pears, quince, and other fruits from the Rosaceae family. These plant diseases caused by bacterial pathogens have arisen as significant challenges in agriculture, threatening global food security and economic stability [1,2,3,4]. In Hungary, Ea was first detected in apple orchards in 1996 [5]; since then, Ea has been detected in different plants and various areas [6,7,8].

Ea infection symptoms include wilting, spots, necrosis, and oozing bacterial exudates, leading to vast losses in fruit production and tree health [9,10]. Conventional control methods involving antibiotics (e.g., streptomycin and gentamicin), chemicals, and cultural practices have limitations, such as developing resistance in bacteria and their harmful effects on beneficial microorganisms [11,12]. There are different mechanisms for how bacteria can develop resistance toward phages [13,14]. The primary mechanisms for resistance in Ea are mutating or masking the targeted receptor [15] or the production of competitive inhibitor molecules, preventing phage–receptor binding [16]. Also, bacteria can protect themselves by hindering DNA injection [17] or the digestion of phage DNA in bacteria by R-M systems [18]. The search for alternative techniques has directed researchers to investigate the potential of bacteriophages as natural enemies of Ea.

Bacteriophages, the viruses that infect and replicate in bacteria, have gained considerable interest as potential biocontrol agents against bacterial pathogens [14,19]. These naturally occurring predators have evolved alongside bacteria and can specifically target and kill pathogenic strains, leaving beneficial microorganisms unharmed [20,21]. Their unique host-specificity and self-replicating properties make bacteriophages attractive candidates for developing targeted and sustainable approaches against Ea infection. Phages are considered environmentally safe options for managing bacterial infections in plants. They offer targeted action against host bacteria, pose no toxicity risks to plants or beneficial microbes, and effectively eliminate antibiotic-resistant bacteria. So, developing strategies using bacteriophages reduces chemical pesticide use while effectively combating bacterial diseases in agriculture [22,23].

Several recent studies have focused on isolating, characterizing, and applying bacteriophages to control Ea infections. These phages exhibit high specificity towards the pathogen, efficiently reducing bacterial populations in laboratory and field conditions [24,25,26]. In Hungary, researchers have successfully isolated bacteriophages that are effective against the bacterium Ea [27,28,29]. Bacteriophage cocktails containing several phages are developed to counteract the emergence of bacterial resistance [30,31]. Cocktail formulations provide a broader spectrum of activity, targeting different strains of Ea, thereby enhancing the efficacy and reliability of phage-based control strategies.

Integrating bacteriophages as sustainable tools to defeat Ea infections could be a more environmentally friendly and economically viable approach. This study focuses on Ea phages, which we isolated in Hungary. We characterized the comprehensive features of phages, such as their stability to biophysical factors, host range, one-step growth curve, genome, and morphology, to consider the application of bacteriophages as a new weapon for biological control of fire blight.

## 2. Materials and Methods

### 2.1. Bacterial Strains

For phage isolation, purification, and characterization, Ea strains were obtained from the National Collection of Agricultural and Industrial Microorganisms (Hungary) and are listed in Table 1. Bacterial strains were grown in an LB broth [32] and on LB 1.5% agar plates.

### 2.2. Isolation and Purification of Bacteriophages

Samples for bacteriophage isolation were collected in Hungary, Pecs, from an apple garden, where fire blight symptoms were seen. A total of 10 samples of apples and soil were taken. The Ea strains CFBP 1430 and Ea 4/82 were used to isolate the phages. Isolation was performed in parallel with these two strains in an LB broth, according to the method provided by Gill et al. [34]. Phage detection was conducted using a spot test and the double-agar overlay method [35,36,37]. Single plaques were taken and resuspended in an SM buffer (50 mM of Tris, 10 mM of NaCl, 8 mM of MgSO_4_, and pH 7.4) then centrifuged and filtered via a 0.22 mm membrane filter. This procedure was repeated 3 times. In order to choose likely new phages, to check the purity of phage suspensions, and to exclude cross-contamination, after every stage of purification, chosen plaques were tested by qPCR [38] with already-existing phage primers, such as Y2, M7, L1, S6, S2, Bue-1, EaH1, Ea H2, and Ea 35–70 [27,28,39,40].

### 2.3. Transmission Electron Microscopy

High-titer phage stocks (10^11^ PFU/mL) were obtained by the double-agar overlay method, scaping the top agar layer, as described by Makalatia Kh et al. [41]. Concentrated phages were analyzed under a JEM-1400 Flash transmission electron microscope (JEOL; Tokyo, Japan), according to the procedure by Koderi Valappil et al. [42].

### 2.4. Plaque Morphology and Quantitative and Qualitative Evaluation of Bacteriophages

To evaluate the phages’ infectivity and morphology, we performed quantitative, double-agar overlay investigations [37] and qualitative spot tests [36]. We characterized the plaque morphology of the phages based on the double-agar method. Infectivity is represented as the efficiency of plating (EOP) [37,43]. EOP was calculated as follows:$$\mathrm{EOP}=\frac{T}{T_{0}}$$
where

T—titer of phage with tested bacteria;

$T_{0}$—titer of phage with host bacteria.

If EOP = 1, the tested bacteria have the same output of phages as the host strain. If EOP > 1, the tested bacteria should be considered a better host strain. If E < 1, the tested bacteria have less phage output than the host strain.

Spot test results were analyzed as follows, according to Kutter E. [36]: 4+—confluent lysis (CL); 3+—semi-confluent lysis (SCL); 2+—confluent lysis with opacity due to secondary growth (OL); 1+—on the bacterial growth, individual lysis sites are small dots (IPOs); and no lysis.

### 2.5. In Vitro Infection Assay

A multi-plate reader assay was used to determine the efficiency of the phages in a liquid medium. Bacterial overnight (ON) cultures were adjusted to an optical density of OD_600_ = 0.5 then were diluted 10X in 1 mL of an LB broth. A total of 160 µL of LB, 20 µL of bacterial suspension, and 20 µL of phages from sufficient dilution were added to 96-well-plate wells to obtain an MOI (Multiplicity of Infection) of 1 to 0.000001. Incubation was performed at 26 °C for 16 h, with medium shaking, and the OD was recorded at 600 nm with a 30 min frequency. Each dilution was prepared in three replicates, and the experiment was repeated three times.

### 2.6. Stability of Bacteriophages

The stability of the phages was tested at different temperatures (50 °C, 60 °C, and 70 °C) for 30 min, 60 min, and 120 min, respectively [44]. A total of 100 µL from each suspension was aliquoted and diluted with serial dilutions. Titer was determined by the double-agar overlay method. 

The stability of the phages was tested at different pHs of 3.0, 4.0, 6.0, 7.0, 8.0, and 9.0 for 24 h [44]. Phages were incubated in an LB broth. The exposure time could vary depending on the specific goals of the experiment, so sampling was performed after 30 min, 60 min, and 24 h. A total of 100 µL from each pH test was aliquoted and diluted with serial dilutions. Titer was determined by the double-agar overlay method. 

The desiccation tolerance of the phages was examined by incubating 10 µL of phage suspensions on glass slides in three technical parallels. Sampling was performed for the following periods: T = 0, T = 30 min, and T = 24 h. After drying, for sample T_0_, 30 μL of an SM buffer was added and mixed well with a pipette tip, and then, 20 μL of it was diluted in a 180 μL SM buffer, and then, additional 10x dilutions were prepared from 10^−2^ to 10^−9^. The same method was used to test slide samples after 30 min and 24 h of incubation.

We selected UV-B radiation to assess the tolerance of phages to UV light. We determined the optimal exposure doses by measuring the irradiation of lamps at various distances. Based on these results, we determined the optimal distance and exposure time to achieve 1, 2, and 4 Joule/centimeter^2^ (J/cm^2^) exposure doses. Phage titer was evaluated by the double-agar overlay method. The exposure dose was calculated as follows:$$Expose\;Dose=Irradiation\left(\frac{Watt}{{cm}^{2}}\right)*Expose\;time\left(seconds\right)$$

The following equation calculated the effect of environmental factors on phage viability:$$Phage\;Viability=\mathrm{Lg}\frac{N_{0}}{N_{t}}$$
where N_0_ is the initial titer of the phage, and N_t_ is phage titer after t time under the effect of physical factors.

### 2.7. One-Step Growth Curve

A bacterial ON culture was diluted in 10 mL of LB, and a sufficient amount of the phages was added to obtain MOI = 0.01, incubated for 10 min, and then centrifuged at 12,000× *g* for 2 min to remove unattached phages. The remaining sediment was re-diluted with 10 mL of an LB broth [45]. The sampling was conducted every 5 min for 20 min and then every 10 min for 120 min. Titer was determined by the double-agar overlay method. 

The latent period refers to the interval between the initial infection of a bacterial cell by phage and the subsequent release of mature phage particles. The burst size measures the number of phage particles released from an infected bacterial cell. It is calculated as the ratio of phage particles released at the plateau level (when the bacterial culture reaches its maximum phage concentration) to the initial number of infected bacterial cells [46]. 

### 2.8. DNA Isolation and Genome Sequencing

According to the manufacturer’s instructions, DNA was isolated from purified phage suspension using a High Pure Viral Nucleic Acid Kit (Roche Diagnostics GmbH, Mannheim, Germany). For DNA sequencing, the DNA library was prepared with a Nextera XT DNA Library Prep Kit (Illumina Inc., San Diego, CA, USA), according to the manufacturer’s instructions. Phage DNA were sequenced by an Illumina (Illumina Inc., San Antonio, TX, USA) with the MiSeq System (v2 Reagent Tray 300 cycles-PE; 2 × 150 bp paired-ends mode). The assembly of sequence data was performed using SPAdes assembler V.3.9.0. ORFs were predicted and genomes annotated by the RAST-Rapid Annotation system using the Subsystem Technology server (https://rast.nmpdr.org (accessed on 20 December 2023)), and genes were predicted using https://proksee.ca/ (accessed on 20 December 2023) by Bakta v1.8.2 (DB: v5.0-Light). Genomes were blasted by Nucleotide Blast-BLASTn (https://blast.ncbi.nlm.nih.gov/Blast.cgi (accessed on 20 December 2023)). Pairwise and multiple sequence alignments were performed using Geneious Prime version 2025.1.2 (Biomatters Ltd., Auckland, New Zealand), employing the MAFFT algorithm (v7.490). Phylogenetic analyses were conducted in Geneious Prime with the Geneious Tree Builder tool, applying the Tamura–Nei genetic distance model and the Neighbor-Joining tree construction method. The following sequences were used for tree construction (GenBank accession numbers): MW349138; PQ051110; OQ818694; OK129343; OQ818696; OQ818698; OQ818701; NC_019504; NC_048875; OQ818704; OQ818695; OQ818699; OQ818700; OQ818697; and OQ818702, PQ431419, and PQ431420.

## 3. Results and Discussion

For the phage sample purification process, isolated plaques with different plaque morphologies were chosen. After every stage of the purification process, chosen plaques were tested with qPCR with already-existing phage primers, such as Y2, M7, L1, S6, S2, Bue-1, EaH1, Ea H2, and Ea 35–70. Based on the qPCR results from the final stage, 12 phages were positive with L1; 4 with EaH1; and 2 phages, EaPF7 and Ea PF9, provided negative results with all primers. Thus, these two phages were identified as probably novel and selected for further characterization, which was conducted with Ea 4/82.

### 3.1. Viral and Plaque Morphology

The phages’ plaque morphology was evaluated on LB agar plates with the host bacteria Ea 4/82. For Ea PF 7, completely transparent plaques with a 2–4 mm diameter were obtained; however, Ea PF 9 showed more uniform plaques with a diameter of 4 mm (Figure 1).

A transmission electron microscopy analysis determined the morphology of the phages. These phages exhibited an icosahedral head with a size of 60 ± 2 nm, and they featured a long, contractile tail measuring 108 ± 2 nm (Figure 1).

### 3.2. Quantitative and Qualitative Evaluation of Bacteriophages

Ea PF 7 and Ea PF 9 phage susceptibility was evaluated against 37 Ea strains from the National Collection of Agricultural and Industrial Microorganisms (Hungary). The lytic activity of the phages was determined with a spot test, while the double-agar method was used for quantitative evaluation (Table 2). The efficiency of each phage was evaluated at different MOIs in a liquid culture, using a 96-well-plate method with an uninterrupted time regime.

The spot test revealed that both the Ea PF 7 and Ea PF 9 phages could lyse 95% of the bacteria tested. However, Ea PF 7 showed a marginally higher efficacy than Ea PF 9. The quality of lysis induced by Ea PF 7 was as follows: a total of 43% of the bacteria exhibited confluent lysis, 40% exhibited semi-confluent lysis, and 12% exhibited opaque lysis. In comparison, Ea PF 9 induced confluent lysis in 40% of the bacteria, semi-confluent lysis in 38%, and opaque lysis in 17%, respectively. These results suggest that Ea PF 7 has slightly higher efficacy in lysing bacterial strains, highlighting the potential of both phages as helpful biocontrol agents.

Furthermore, the efficiency of plating (EOP) was calculated to be more than 0.7 in 86.5% of Ea strains for Ea PF 7 and 81% for Ea PF 9 compared to a host strain, which indicates that the phages can replicate and produce progeny phages efficiently.

The lytic activity of a bacteriophage, which refers to its ability to infect and destroy bacterial cells, is a critical factor in determining its prospect for various applications. In this study, the lytic activity of each phage was investigated at extremely low MOIs (MOI of 0.0000001), and in both cases, a significant inhibition of bacterial growth for nearly 15 h was observed (Figure 2a,b). The lytic activity of these phages was remarkable, even at deficient concentrations. This ultralow effective MOI could significantly reduce the costs associated with the practical application of the phages [47]. The development of phage-resistant mutants was observed in Ea bacteria, which is notable in the context of phage–bacteria interactions, as reported in previous studies [31,32]. Bacterial growth was detected after 15 h of phage exposure, suggesting the potential emergence of resistant mutants. To attribute the results obtained from the multi-well plate reader to the emergence of phage-resistant bacterial mutants, double-agar plate assays were performed. After 48 h of incubation, mutant colonies were observed and isolated. Their resistance was confirmed by spot testing [48].

The latent period of the phages Ea PF 7 and Ea PF 9 was short, lasting only 10 min (Figure 2c,d). The burst size, which refers to the number of phage particles released from a single infected bacterial cell, for Ea PF 7 and Ea PF 9 was 108 and 125 phage particles per infected cell. The combination of a short latent period and a high burst size, characteristic of highly effective lytic phages [49], likely explains the exceptional results observed at extremely low MOIs. These characteristics enable efficient phage propagation even when the initial phage concentration is minimal, ensuring rapid infection cycles and high phage yields relative to the available host population. These features make Ea PF 7 and Ea PF 9 particularly suitable for further phage cocktail formulations to combat bacterial infections, especially in environments where such traits are critical for efficiently controlling bacterial diseases.

### 3.3. Stability to Biophysical Factors

Ea PF7 and Ea PF9 exhibited remarkable stability at moderate temperatures (37 °C and 45 °C), with no significant changes observed in their titers after a 120 min incubation. However, exposure to higher temperatures (50 °C to 55 °C) rapidly reduced their titer. Ea PF7 and Ea PF9 could not survive at 60 °C (Figure 3). In the case of Ea PF9, the data reveal that after 90 min of incubation, the phage titer dropped to 0, indicating complete inactivation. This outcome aligns with the rapid loss of viability observed at higher temperatures, particularly at 55 °C, as shown in Figure 3b.

The pH stability test revealed that both Ea PF7 and Ea PF9 can tolerate a wide range of pH conditions, from pH 4 to pH 10, after a 24 h incubation period. At the pH values pH 4 and pH 10, a notable reduction in titer by 2–3 logs was observed, indicating a loss of infectivity (Figure 4).

Our results also demonstrate that both phages show considerable desiccation tolerance, as evidenced by minimal changes in the titer after a 24 h drying period, Figure 5a. This is an important finding, as desiccation tolerance is crucial for the potential use of phages in various applications, including as therapeutics or in biotechnology processes that require long-term survival.

The irradiation dose is determined by several factors, including wavelength, lamp power (as specified by the manufacturer), distance from the lamp, exposure duration, and the age of the lamp, which impacts its power output over time. Experimental studies often omit specific parameters, resulting in incomplete datasets and outcomes that lack comparability and reproducibility. For instance, some studies report only the wavelength and exposure duration, omitting the lamp power and distance [50,51,52], while others provide the lamp power and duration but exclude details on the wavelength and distance [53]. Even when all parameters are given [54,55], the gradual decline in lamp power due to aging, introduces variability in the irradiation dose, even with the same lamp.

To standardize experimental conditions, this study recommends measuring the lamp intensity (W/cm^2^) at a defined distance at the time of use and calculating the precise irradiation dose (J/cm^2^) based on the exposure duration. This method allows for the accurate assessment of irradiation effects on phage viability and ensures the reproducibility of experimental results.

At UV doses of 1 and 2 J/cm^2^, the phages demonstrated a relatively high degree of stability, with a decrease in viability of only 0.4 to 0.9 logs. This suggests that these phages can tolerate UV radiation at these lower doses. However, at a higher UV dose of 4 J/cm^2^, the viability of the phages decreased significantly, with a reduction in viability of 1.6 to 1.9 logs (Figure 5b). The phages exhibited notable desiccation, temperature, and pH tolerance, as evidenced by minimal changes in relevant parameters close to natural environmental conditions. Phages are more sensitive to UV radiation at this higher dose, consistent with previous studies showing that UV radiation can damage the genetic material of phages, leading to a loss of infectivity [56].

### 3.4. Genomic Analysis

Ea PF7 and Ea PF9 had 54,114 bps and 52,943 long genomes with a 43.9% and 44.0% G + C content, respectively. The DNA sequencing coverage achieved was 66X for Ea PF7 and 216X for Ea PF9. In the genome of Ea PF7, 87, and in Ea PF9, 84, open reading frames (ORF) were detected. The function of most ORFs could not be annotated, so they were defined as hypothetical proteins, but there are also 33 functional ones in both phage genomes. Based on annotation, structural proteins, proteins related to packaging, cell lysis proteins, and nucleic acid metabolism proteins were defined. However, no proteins associated with lysogeny, such as C1 repressor-like proteins [57], were identified, nor were any antibiotic resistance genes detected. This absence enhances its suitability as a biocontrol agent.

The phage genomes were analyzed using pairwise alignment, revealing a high degree of similarity, approximately 95.6%, in their genomic sequences. As the difference in genomes is 4.4%, only the whole genomic map of Ea PF 7 is presented in Figure 6. Despite this, they exhibited notable differences in other phenotypic characteristics during additional testing. Genomic differences were primarily located in non-coding regions or regions encoding hypothetical proteins, with the exception of the homing endonuclease genes, which exhibited 48.0% identity in pairwise alignment. Homing endionucleases represent an evolutionary strategy that phages use to ensure that their genetic elements persist and spread, while potentially providing benefits in phage–host competition and adaptability [58]. Phages with closely related genomes but distinct host ranges often have subtle genetic differences in genes coding for receptor-binding proteins or tail fibers [59]. While the annotation did not specifically identify these proteins, it is plausible that they are encoded by hypothetical proteins, which may conceal their functional roles.

According to the International Committee on Taxonomy of Viruses (ICTV), the genome analysis determined that both phages are Loessnervirus-like viruses. Genome blast results demonstrate that Phi Fifi044, Phi Fifi451 [50], and SNAUBM-27 [31] from Korea were the most related bacteriophages to Ea PF 7 and Ea PF 9.

Seventeen complete genomic sequences of Loessnervirus-like phages were used to construct phylogenetic trees, including the genomes of Ea PF 7 and Ea PF 9. Two distinct trees were generated: the first was based on concatenated sequences of the portal protein and terminase large subunit genes (Figure 7a), while the second utilized whole-genome sequences (Figure 7b). In the concatenated gene-based phylogeny, Ea PF 7 (GenBank accession no. PQ431420) and Ea PF 9 (PQ431419) clustered with OQ818694 and several other phages encoding putative portal proteins (MW349138, PQ051110, and OK129343) (Figure 7a). In contrast, the whole-genome-based phylogeny placed EaPf7 and EaPf9 in a distinct clade, separate from OQ818694 (Figure 7b). This topological incongruence may reflect a recombination or horizontal gene transfer event involving structural module genes, aligning with the modular evolution model of bacteriophages.

Admitting that the same phages have also been isolated in Hungary contributes to the broader knowledge of their geographical distribution and relevance. This not only contributes to understanding their distribution but also raises questions about the genetic variability and adaptability of these phages in different environmental and geographical contexts.

## 4. Conclusions

This study demonstrated that the phages Ea PF 7 and Ea PF 9 exhibited strong efficacy and stability under environmental conditions, characterized by a short latent period and large burst size. Genome sequencing revealed no genes associated with lysogeny or antibiotic resistance. While minor differences were observed between the two genomes, their biological features remained distinct. These findings position Ea PF 7 and Ea PF 9 as promising candidates for further cocktail formulations, which will be the focus of our future research. Developing bacteriophage-based biocontrol agents holds great potential for combating plant infectious diseases.

## Figures and Tables

**Figure 1 viruses-17-00677-f001:**
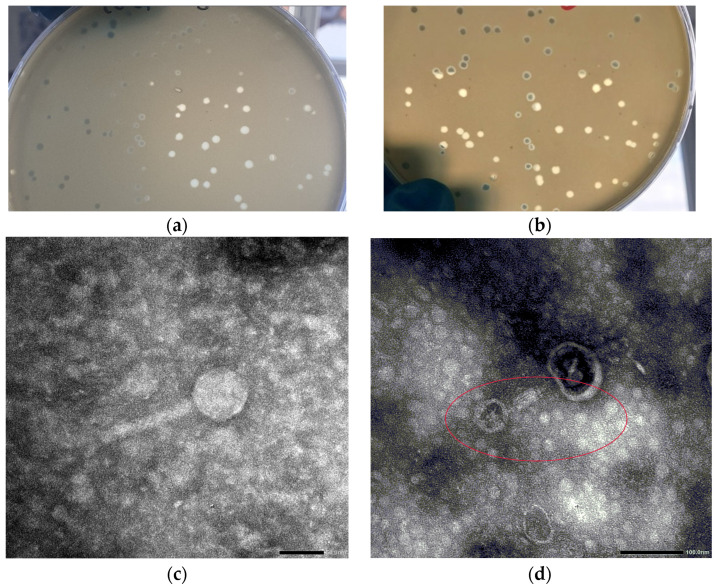
Photos of the plaques of the phages Ea PF 7 (**a**) and Ea PF9 (**b**). Transmission electron micrographs of phage Ea PF 7; scale bar = 50 nm (**c**). Transmission electron micrographs of phage Ea PF 7 contracted position; scale bar = 100 nm. Red circle highlight theEaPf7 bacteriophage with a contracted tail. (**d**).

**Figure 2 viruses-17-00677-f002:**
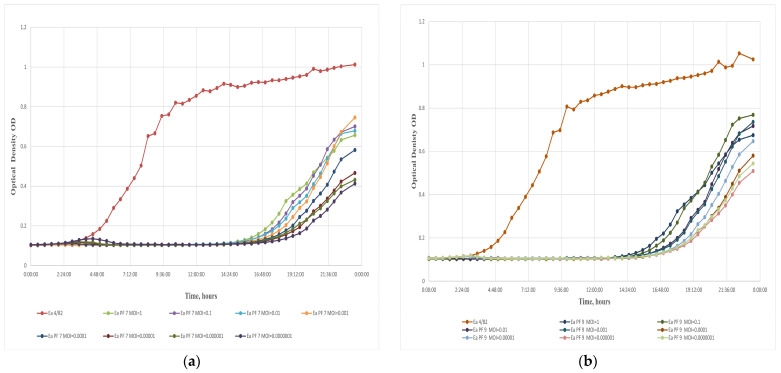
Effect of the bacteriophages on host bacteria growth. Evaluation of phage activity at different MOIs for phage Ea PF 7 (**a**) and for phage Ea PF 9 (**b**). One-step growth curve of Ea PF 7 (**c**) and of Ea PF 9 (**d**).

**Figure 3 viruses-17-00677-f003:**
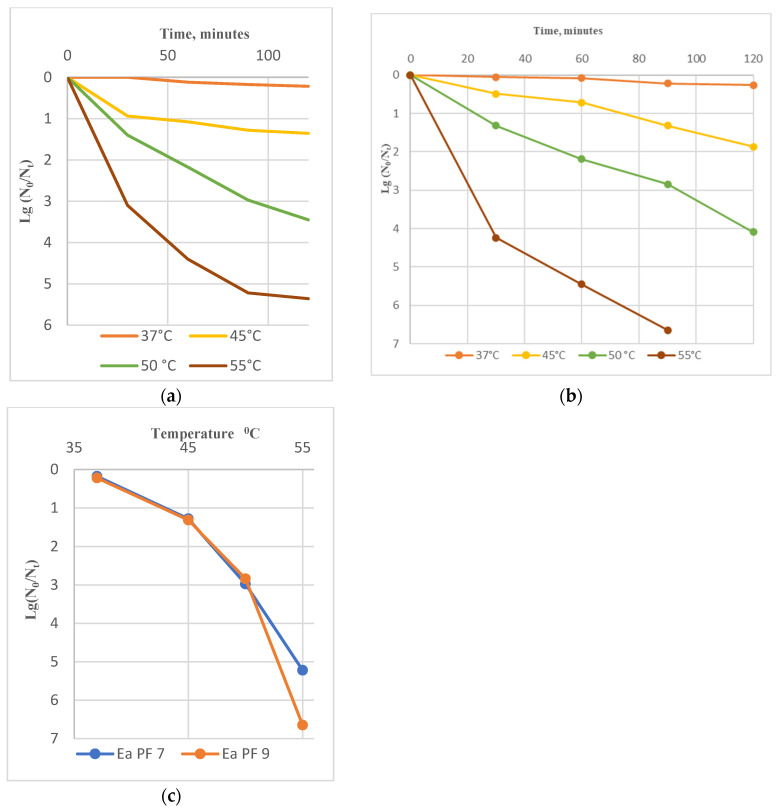
Temperature stability of phage Ea PF 7 (**a**) and Ea PF 9 (**b**). Comparative characterization of phages’ stability during 90 min incubation at different temperatures (**c**).

**Figure 4 viruses-17-00677-f004:**
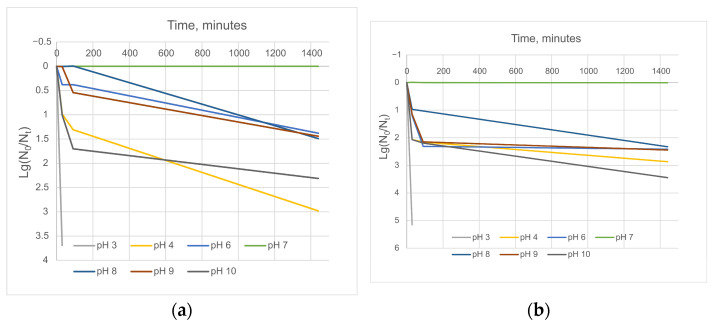
pH stability of phage Ea PF 7 (**a**) and Ea PF 9 (**b**). Comparative characterization of phages’ stability during 30 min incubation at different pHs (**c**).

**Figure 5 viruses-17-00677-f005:**
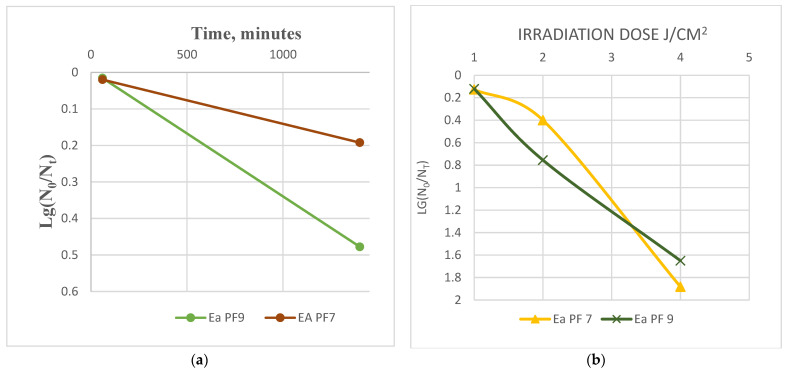
Desiccation tolerance of the phages (**a**). UV-B tolerance of the phages (**b**).

**Figure 6 viruses-17-00677-f006:**
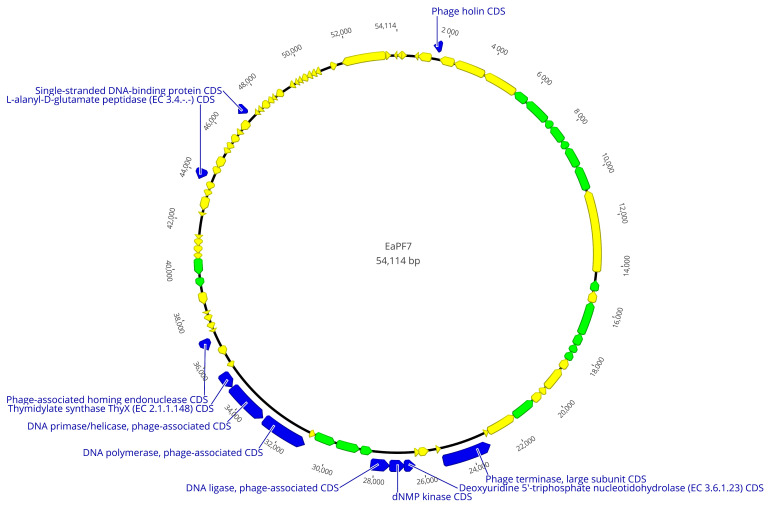
Genomic maps of phage Ea PF 7. Blue label—functional proteins; green label—structural proteins; yellow label—hypothetical proteins.

**Figure 7 viruses-17-00677-f007:**
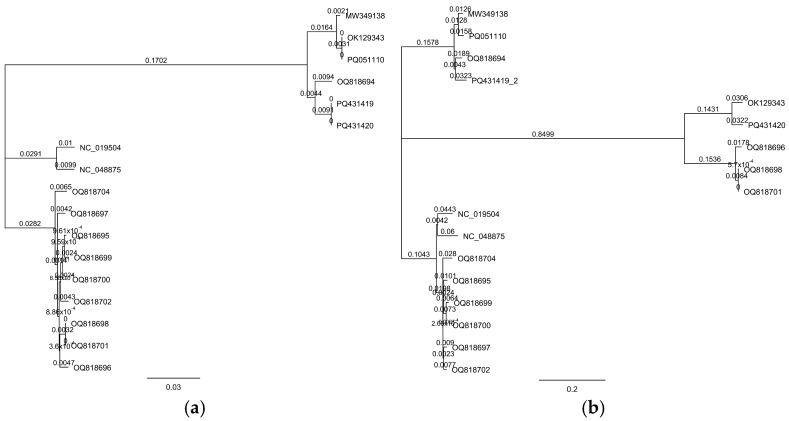
Phylogenetic trees of Ea PF 7 (GenBank accession no. PQ431420) and Ea PF 9 (PQ431419). (**a**) Tree constructed using concatenated amino acid sequences of the portal protein and terminase large subunit genes. (**b**) Tree constructed using complete whole-genome sequences.

**Table 1 viruses-17-00677-t001:** List of the Ea strains used in this study.

Ea	Strain Collection and Origin	Isolation Source
Ea B 01272	National Collection of Agricultural and Industrial Microorganisms (Hungary)	Apple
Ea B 01616	“	Unknown (plant)
Ea B 01728	“	Apple
Ea B 01729	“	Apple
Ea B 01731	“	Apple
Ea B 01733	“	Apple
Ea B 01734	“	Pear
Ea B 01735	“	Quince
Ea B 01738	“	Medlar
Ea B 01756	“	Apple
Ea B 01757	“	Apple
Ea B 01840	“	*Crataegus* sp.
Ea B 01843	“	Apple
Ea B 01844	“	Pear
Ea B 01853	“	*Cotoneaster* sp.
Ea B 01855	“	*Pyracantha* sp.
Ea B 01896	“	Unknown (plant)
Ea B 01898	“	Apple
Ea B 01901	“	Apple
Ea B 01903	“	Apple
Ea B 01905	“	Apple
Ea B 01906	“	Apple
Ea B 01960	“	Apple
Ea B 01961	“	Apple
Ea B 01962	“	Apple
Ea B 01963	“	Apple
Ea B 01964	“	Apple
Ea B 01971	“	Apple
Ea B 01972	“	Pear
Ea B 01973	“	Pear
Ea B 01975	“	Pear
Ea B 01978	“	Pear
Ea B 01980	“	Apple
Ea B 01983	“	Quince
Ea G254	Government Office of Baranya Country, Hungary	Apple
Ea G255	Government Office of Baranya Country, Hungary	Apple
Ea B 0118T	National Collection of Agricultural and Industrial Microorganisms (Hungary)	Pear
CFBP 1430	CIRM-CFBP French Collection for Plant-Associated Bacteria	*Crataegus* sp.
Ea 4/82	Egypt, 1982 [33]	Pear

**Table 2 viruses-17-00677-t002:** Results of qualitative and quantitate evaluation of phages. Lysis activity was analyzed as follows: 4—confluent lysis (CL); 3—semi-confluent lysis (SCL); 2—confluent lysis with opacity due to secondary growth (OL); 1—on the bacterial growth, individual lysis sites are as small dots (IPOs); and 0—no lysis. EOP was analyzed as follows: if EOP = 1, the tested bacteria have the same output of phages as the host strain; if EOP > 1, the tested bacteria should be considered a better host strain; and if EOP < 1, the tested bacteria have less phage output than the host strain.

Ea Strains	Lysis Activity		EOP	
	Ea PF 7	Ea PF 9	Ea PF 7	Ea PF 9
Ea B 01272	3	0	0.5	0
Ea B 01616	2	3	0.7	0.8
Ea B 01728	4	3	0.7	0.7
Ea B 01729	4	1	0.7	0.3
Ea B 01731	2	1	1	0.1
Ea B 01733	4	2	1	0.4
Ea B 01734	4	4	0.9	0.9
Ea B 01735	4	4	0.9	0.8
Ea B 01738	4	3	0.9	0.9
Ea B 01756	4	4	1	1
Ea B 01757	4	4	1	1.1
Ea B 01840	4	4	1	0.9
Ea B 01843	3	3	0.6	0.2
Ea B 01844	3	3	0.6	0.5
Ea B 01853	3	4	0.7	0.8
Ea B 01855	3	3	0.7	0.8
Ea B 01896	2	3	0.7	0.9
Ea B 01898	3	3	0.7	0.4
Ea B 01901	4	3	1.1	1
Ea B 01903	3	3	0.9	0.9
Ea B 01905	3	3	0.9	1
Ea B 01906	4	2	1.2	0.7
Ea B 01960	3	4	0.8	1.1
Ea B 01961	2	2	0.7	0.7
Ea B 01962	4	4	1	0.9
Ea B 01963	4	4	1	1
Ea B 01964	4	4	0.9	1.1
Ea B 01971	3	4	0.7	0.9
Ea B 01972	3	3	0.8	0.8
Ea B 01973	4	4	0.9	1.1
Ea B 01975	3	3	0.7	0.9
Ea B 01978	4	3	0.8	0.7
Ea B 01980	1	4	0.2	1.1
Ea B 01983	3	3	1	0.9
Ea G254	3	4	0.8	1
Ea G255	3	4	0.8	1
Ea B 0118T	0	2	0	1.1
CFBP 1430	3	3	0.8	0.7
Ea 4/82	4	4	1	1

## Data Availability

We submitted assembled and annotated genomes to the GenBank, and the following accession numbers were used: PQ431420 for phage Ea PF 7 and PQ431419 for phage Ea PF 9.

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
