# Peer review of "Characterizations of Newly Isolated Erwinia amylovora Loessnervirus-like Bacteriophages from Hungary"

_viruses, 2025, doi:10.3390/v17050677_

Round 1
Reviewer 1 Report
Comments and Suggestions for Authors
In this manuscript Elene Lomadze and colleagues present isolation and functional characterization of two new phages infecting plant pathogenic bacterium Erwinia amylovora. The new phages potentially could be used for biocontrol of this disease. The study is technically sound and generally well presented. The main concern is the rather scant and uninformed Discussion section, which mostly repeats and summarizes the results.
Specific comments.
line 16. Burst size should be specified.
lines 19-21. I do not understand this sentence. Apparently UV-tolerance should be evaluated under controlled conditions. What is novel?
Line 151 and table 2. Why it was important to check PCR amplification with already published primers? You isolated new phages and it is quite unlikely that they will match previous primers. Table 2 show only negative results and thus is unnecessary.
Lines 158-159. Isolation of phages Ea PF 7 and Ea PF 9 should be described here. What was the source, how they were isolated?
Figure 2 Panels A and B are of poor quality, better resolution is required
Figure 3 and 4. What is shown in panel C? Should be specified in figure legend
Figure 6 is of poor quality. Since the genomes are 98% identical, it is enough to show one of them.
More detail genome analysis should be presented. For example, what was “functional proteins”, did you found genes relevant to lysogeny etc.
Discussion essentially repeats the Results. I would recommend here to compare the new phages with other known to infect Erwinia amylovora.
Author Response
Comments and Suggestions |
Lines is old version |
Answer |
Lines in new corrected version |
Referee 1 |
|||
Burst size should be specified |
Line 16
|
Burst size is included |
Line 17 |
I do not understand this sentence. Apparently, UV-tolerance should be evaluated under controlled conditions. What is novel? |
Lines 19-21 |
Sentense is corrected.
Expanation about UV test is also expended in manuscript. Answer: Irradiation dose depends on many factors such as wavelength, lamp power according to manufacture, distance from the lamp, irradiation time, and the lamp’s age because the lamp power will change. In many experiments, not all factors are considered, and some aspects are missed, which makes results not comparable because of missing parameters; they will not be repeated. In some examinations, it is given only wavelength and Time, no lamp power, no distance (Park J, 2018. Luo et al., 2023), or is given only lamp power and Time and no distance and no wavelength (Jo SJ et al., 2023) is the wavelength, Time, lamp power according to manufacture, distance (Born Y et al., 2015, Fu J, 2023). However, because of the aging of the lamp, power is still decreasing. Irradiation dose may vary, even if you use the same lamp, and after some extended period, power will be less. So, we offer to measure lamp intensity (Watt/cm2) at a chosen distance, at a given moment, and the variable time of exposure, and calculate the exact exposed irradiation doses (J/ cm2) and its effect on phage viability. Park J, Lee GM, Kim D, Park DH, Oh CS. Characterization of the Lytic Bacteriophage phiEaP-8 Effective against Both Erwinia amylovora and Erwinia pyrifoliae Causing Severe Diseases in Apple and Pear. Plant Pathol J. 2018 Oct;34(5):445-450. doi: 10.5423/PPJ.NT.06.2018.0100. Epub 2018 Oct 1. PMID: 30369854; PMCID: PMC6200048. Luo J, Xie L, Yang M, Liu M, Li Q, Wang P, Fan J, Jin J, Luo C. Synergistic Antibacterial Effect of Phage pB3074 in Combination with Antibiotics Targeting Cell Wall against Multidrug-Resistant Acinetobacter baumannii In Vitro and Ex Vivo. Microbiol Spectr. 2023 Aug 17;11(4):e0034123. doi: 10.1128/spectrum.00341-23. Jo SJ, Kim SG, Park J, Lee YM, Giri SS, Lee SB, Jung WJ, Hwang MH, Park JH, Roh E, Park SC. Optimizing the formulation of Erwinia bacteriophages for improved UV stability and adsorption on apple leaves. Heliyon. 2023 Nov 8;9(11):e22034. doi: 10.1016/j.heliyon.2023.e22034 Born Y, Bosshard L, Duffy B, Loessner MJ, Fieseler L. Protection of Erwinia amylovora bacteriophage Y2 from UV-induced damage by natural compounds. Bacteriophage. 2015 Jul 24;5(4):e1074330. doi: 10.1080/21597081.2015.1074330. Fu J, Li Y, Zhao L, Wu C and He Z.Characterization of vB_ValM_PVA8, a broad-host-range bacteriophage infecting Vibrio alginolyticus and Vibrio parahaemolyticus. Front. Microbiol. 2023; 14:1105924. doi: 10.3389/fmicb.2023.1105924.
|
Lines 23.
UV explanation: Lines -266-279 |
Why it was important to check PCR amplification with already published primers? You isolated new phages and it is quite unlikely that they will match previous primers. Table 2 show only negative results and thus is unnecessary.
|
Line 151 and table 2 |
About PCR corected in manuscript, table is deleted.
Answer: During isolation process, when we started picking up isolated plaques, we have no idea they were knew phages or not, so to continue with probably new ones we tested it with PCR. Because if there were positive results, we will not choose such phages for farther characterizations. as well as to check purity of the phages and avoid cross-contamination.
|
Lines 86-89 |
Isolation of phages Ea PF 7 and Ea PF 9 should be described here. What was the source, how they were isolated?
|
Lines 158-159 |
Information is added in manuscript |
Lines 79-86 |
Figure 2 Panels A and B are of poor quality, better resolution is required |
|
Figures changed |
|
Figure 3 and 4. What is shown in panel C? Should be specified in figure legend |
|
Corrected in manuscript |
|
Figure 6 is of poor quality. Since the genomes are 98% identical, it is enough to show one of them.
|
|
Corrected, Only figure of Ea PF7 is presented now |
|
More detail genome analysis should be presented. For example, what was “functional proteins”, did you find genes relevant to lysogeny etc. |
|
Finctinal proteins are more specified in manuscript. |
Lines 297-303 |
Discussion essentially repeats the Results. I would recommend here to compare the new phages with other known to infect Erwinia amylovora. |
|
Changed style of discussion
|
|

Reviewer 2 Report
Comments and Suggestions for Authors
Referee Report
Lomadze, E et al.
Characterization of Newly Isolated Erwinia amylovora Loessnervirus-like Bacteriophages from Hungary.
In this paper the authors describe the genetic and physiological features of two different Erwinia phage, grouped into the Loessnervirus section. A primary aim is to find treatments against the plant fireblight caused by Erwinia bacteria.
The experiments analyse the plaque morphology, burst size and Erwinia-inhibiting potential of the two phage. In addition, genome analyses are shown.
The isolation and characterization of new phages, particulary those which attack plant pathogenic bacterial strains is an important and interesting topic in food-associated processes and in orchards as it is not recommendable to apply antibiotics there. However, for Erwinia species there have already numerous phages been characterized attacking these pathogens, so that the novelty of this paper is limited.
The paper touches a number of aspects of the biology and genetics of the new phages. However, some experimental approaches have to be redesigned to draw the conclusions the authors made.
Table 2:
This table only shows negative data and is therefore not really helpful. Please add positive controls for the various phage tested. Why does the title of the table say Ea PF8?
Table 3:
What is the reference strain for the EOP data? Plating on Ea4/82?
Figure 1:
The TEM picture does not give enough information to draw the conclusion that EaPF7 has a contractile tail. The authors have to show more than only 1 phage particle and specifically compare individuals with contracted vs. non-contracted tails. What is the size of the scale bar?
Fig. 2
This figure (a and b) shows growth curves of the infected bacteria together with an uninfected control. Basically, phage infection only leads to a growth delay and probably the cultures reach the same OD after 2 more hours. The authors do not give any explanation why the growth delay is rather similar no matter whether an MOI of 1 or 10^-7 is used. This is rather strange as it is not understandable that at these low MOIs (only one phage particle per 1-10 million of bacteria) the culture growth is inhibited substantially. The authors have to use fresh cultures rather than diluted overnight cultures. A lysis curve has to be shown. The data are contradictory to the statement in fig. 2 c and d that the phage have extremely short latent times.
Fig 2 c and d are stated to show one step growth curves. This is not the case. No bacterial counts are given and there are obviously two steps in the phage titer curves. One-step-growth/infection curves cannot be achieved at an MOI of 0,01.
Fig. 3
In the legend, Fig 3c is missing. How were the N0/Nt numbers calculated? It is not possible to calculate reliable phage titers only from spot tests!
Fig. 4
Same problems as for Fig. 3. In the legend, Fig 4c is missing. How were the N0/Nt numbers calculated? It is not possible to calculate reliable phage titers only from spot tests!
Fig. 5
It is not possible to calculate reliable phage titers only from spot tests! Apply plating assays.
Some spelling errors have to be corrected
· P3, line 74: Transmission Electron Microscopy
line 78: Quantitatively
· P4, line 147 Nucleotide Blast
· Check the uniformity of the phage names: sometimes it is Ea PF7 and Ea PF9, sometimes EA Pf7 and EA PF9 resp.
· Check the word phage titer instead of phage titter

Author Response
Referee 3 |
||||
However, for Erwinia species there have already numerous phages been characterized attacking these pathogens, so that the novelty of this paper is limited.
|
|
Based on the DNA blasting our phages are related only with Korean phage Fifi 044, Fifi 451 (Park J. et al.), and SNAUB 27( Kim SG et al.) Besides the main aim of the study to have well-characterized phages for further usage as an anti-bacterial biocontrol agent, by publishing this work, we aim to formally recognize that the same phages have also been isolated in Hungary, contributing to the broader knowledge of their geographical distribution and relevance. Park J, Kim B, Song S, Lee YW, Roh E. Isolation of Nine Bacteriophages Shown Effective against Erwinia amylovora in Korea. Plant Pathol J. 2022 Jun;38(3):248-253. doi: 10.5423/PPJ.NT.11.2021.0172. Epub 2022 Jun 1. PMID: 35678058; PMCID: PMC9343912. Kim SG, Lee SB, Jo SJ, Cho K, Park JK, Kwon J, Giri SS, Kim SW, Kang JW, Jung WJ, Lee YM, Roh E, Park SC. Phage Cocktail in Combination with Kasugamycin as a Potential Treatment for Fire Blight Caused by Erwinia amylovora. Antibiotics (Basel). 2022 Nov 6;11(11):1566. doi: 10.3390/antibiotics11111566. PMID: 36358221; PMCID: PMC9686651.
|
319-325 |
|
However, some experimental approaches have to be redesigned to draw the conclusions the authors made. |
|
We repetied some stability test with Double agar overlay as well.
|
|
|
Table 2: This table only shows negative data and is therefore not really helpful. Please add positive controls for the various phage tested. Why does the title of the table say Ea PF8?
|
|
Table is deleted |
|
|
Table 3: What is the reference strain for the EOP data? Plating on Ea4/82?
|
|
Refference stain was Ea 4/82, added in manuscript |
Line 176 |
|
Figure 1: The TEM picture does not give enough information to draw the conclusion that EaPF7 has a contractile tail. The authors have to show more than only 1 phage particle and specifically compare individuals with contracted vs. non-contracted tails. What is the size of the scale bar?
|
|
Because of bad quality we do not include TEM photo of phages in contravted position . Now we added it. |
|
|
Fig. 2 This figure (a and b) shows growth curves of the infected bacteria together with an uninfected control. Basically, phage infection only leads to a growth delay and probably the cultures reach the same OD after 2 more hours. The authors do not give any explanation why the growth delay is rather similar no matter whether an MOI of 1 or 10^-7 is used. This is rather strange as it is not understandable that at these low MOIs (only one phage particle per 1-10 million of bacteria) the culture growth is inhibited substantially (answered) .
|
|
Peng et al. investigated the effects of different MOIs (Multiplicity of Infection) on bacterial growth and observed that even at a very low MOI of 0.001, bacterial growth was completely inhibited during 6 hours. Building on these findings, we decided to explore even lower, more extreme MOIs to identify the threshold at which phages are no longer effective in suppressing bacterial growth. This approach aims to pinpoint the optimal MOI for the phages under investigation. short latent period combined with a large burst size likely explains the exceptional results observed at extremely low MOIs. These characteristics enable efficient phage propagation even when the initial phage concentration is minimal, ensuring rapid infection cycles and high phage yields relative to the available host population.
Peng Q, Ma Z, Han Q, Xiang F, Wang L, Zhang Y, Zhao Y, Li J, Xian Y, Yuan Y. Characterization of bacteriophage vB_KleM_KB2 possessing high control ability to pathogenic Klebsiella pneumoniae. Sci Rep. 2023 Jun 17;13(1):9815. doi: 10.1038/s41598-023-37065-5. PMID: 37330608; PMCID: PMC10276810.
|
|
|
The authors have to use fresh cultures rather than diluted overnight cultures. A lysis curve has to be shown.
|
|
Our overnight cultures were prepared by inoculating a single colony of E. amylovora from LB agar plates into 5 mL of LB broth, followed by incubation for 15 hours. This ensured that the bacteria were viable and active, as confirmed by their robust growth in subsequent turbidity and phage assays. Bacterial culture approximately 108 CFU/ml in this case was used. The use of overnight cultures for Erwinia amylovora in similar assays, including turbidity measurements was done by Knecht et al, Savbri et al.,
Knecht LE, Born Y, Pelludat C, Pothier JF, Smits THM, Loessner MJ, Fieseler L. Spontaneous Resistance of Erwinia amylovora Against Bacteriophage Y2 Affects Infectivity of Multiple Phages. Front Microbiol. 2022 Aug 1;13:908346. doi: 10.3389/fmicb.2022.908346. PMID: 35979490; PMCID: PMC9376448.
Sabri M, El Handi K, Valentini F, De Stradis A, Achbani EH, Benkirane R, Resch G, Elbeaino T. Identification and Characterization of Erwinia Phage IT22: A New Bacteriophage-Based Biocontrol against Erwinia amylovora. Viruses. 2022 Nov 5;14(11):2455. doi: 10.3390/v14112455. PMID: 36366553; PMCID: PMC9698647.
|
|
|
The data are contradictory to the statement in fig. 2 c and d that the phage have extremely short latent times(answered) .
|
|
The latent period data varies between 10, 20, and 30 minutes or more, depending on specific conditions.
10 minutes - Peng Q, Ma Z, Han Q, Xiang F, Wang L, Zhang Y, Zhao Y, Li J, Xian Y, Yuan Y. Characterization of bacteriophage vB_KleM_KB2 possessing high control ability to pathogenic Klebsiella pneumoniae. Sci Rep. 2023 Jun 17;13(1):9815. doi: 10.1038/s41598-023-37065-5. PMID: 37330608; PMCID: PMC10276810.
10 minutes - Jo SJ, Kim SG, Lee YM, Giri SS, Kang JW, Lee SB, Jung WJ, Hwang MH, Park J, Cheng C, Roh E, Park SC. Evaluation of the Antimicrobial Potential and Characterization of Novel T7-Like Erwinia Bacteriophages. Biology (Basel). 2023 Jan 23;12(2):180. doi: 10.3390/biology12020180. PMID: 36829459; PMCID: PMC9953017.
Less than 10 minutes - Tang M, Huang Z, Zhang X, Kong J, Zhou B, Han Y, Zhang Y, Chen L, Zhou T. Phage resistance formation and fitness costs of hypervirulent Klebsiella pneumoniae mediated by K2 capsule-specific phage and the corresponding mechanisms. Front Microbiol. 2023 Jul 19;14:1156292. doi: 10.3389/fmicb.2023.1156292. PMID: 37538841; PMCID: PMC10394836.
20 minutes - Fu J, Li Y, Zhao L, Wu C, He Z. Characterization of vB_ValM_PVA8, a broad-host-range bacteriophage infecting Vibrio alginolyticus and Vibrio parahaemolyticus. Front Microbiol. 2023 May 12;14:1105924. doi: 10.3389/fmicb.2023.1105924. PMID: 37250064; PMCID: PMC10213691.
35 minutes - Eskenazi, A., Lood, C., Wubbolts, J. et al. Combination of pre-adapted bacteriophage therapy and antibiotics for treatment of fracture-related infection due to pandrug-resistant Klebsiella pneumoniae. Nat Commun 13, 302 (2022). https://doi.org/10.1038/s41467-021-27656-z
30 minutes - Zurabov F, Zhilenkov E. Characterization of four virulent Klebsiella pneumoniae bacteriophages, and evaluation of their potential use in complex phage preparation. Virol J. 2021 Jan 6;18(1):9. doi: 10.1186/s12985-020-01485-w. PMID: 33407669; PMCID: PMC7789013.
|
|
|
Fig 2 c and d are stated to show one step growth curves. This is not the case. No bacterial counts are given and there are obviously two steps in the phage titer curves.
|
|
We consider that the 10-minute time gap is too short to reliably represent a two-step curve. The experiment was performed in triplicates and repeated three times using the double agar overlay method, sampling every 10 minutes. The data points that deviate from the expected trend in the graph are likely due to measurement inaccuracies. To address this, we have included error bars in the graphs for better representation of the variability. |
|
|
One-step- growth/infection curves cannot be achieved at an MOI of 0,01.
|
|
For our one-step growth curve experiments, we exclusively used a MOI of 0.01 to maintain consistency and ensure reproducibility. While it is common practice in the literature to conduct such experiments with MOIs ranging from 1.0 to 0.0001
MOI = 1 - Peng Q, Ma Z, Han Q, Xiang F, Wang L, Zhang Y, Zhao Y, Li J, Xian Y, Yuan Y. Characterization of bacteriophage vB_KleM_KB2 possessing high control ability to pathogenic Klebsiella pneumoniae. Sci Rep. 2023 Jun 17;13(1):9815. doi: 10.1038/s41598-023-37065-5. PMID: 37330608; PMCID: PMC10276810.
MOI =0.1 - Fu J, Li Y, Zhao L, Wu C, He Z. Characterization of vB_ValM_PVA8, a broad-host-range bacteriophage infecting Vibrio alginolyticus and Vibrio parahaemolyticus. Front Microbiol. 2023 May 12;14:1105924. doi: 10.3389/fmicb.2023.1105924. PMID: 37250064; PMCID: PMC10213691.
MOI = 0.01 - Zurabov F, Zhilenkov E. Characterization of four virulent Klebsiella pneumoniae bacteriophages, and evaluation of their potential use in complex phage preparation. Virol J. 2021 Jan 6;18(1):9. doi: 10.1186/s12985-020-01485-w. PMID: 33407669; PMCID: PMC7789013. MOI= 0.01- Pertics BZ, Cox A, Nyúl A, Szamek N, Kovács T, Schneider G. Isolation and Characterization of a Novel Lytic Bacteriophage against the K2 Capsule-Expressing Hypervirulent Klebsiella pneumoniae Strain 52145, and Identification of Its Functional Depolymerase. Microorganisms. 2021 Mar 21;9(3):650. doi: 10.3390/microorganisms9030650. PMID: 33801047; PMCID: PMC8003838.
MOI-0.001 - Jo SJ, Kim SG, Lee YM, Giri SS, Kang JW, Lee SB, Jung WJ, Hwang MH, Park J, Cheng C, Roh E, Park SC. Evaluation of the Antimicrobial Potential and Characterization of Novel T7-Like Erwinia Bacteriophages. Biology (Basel). 2023 Jan 23;12(2):180. doi: 10.3390/biology12020180. PMID: 36829459; PMCID: PMC9953017.
MOI=0.0001- Tang M, Huang Z, Zhang X, Kong J, Zhou B, Han Y, Zhang Y, Chen L, Zhou T. Phage resistance formation and fitness costs of hypervirulent Klebsiella pneumoniae mediated by K2 capsule-specific phage and the corresponding mechanisms. Front Microbiol. 2023 Jul 19;14:1156292. doi: 10.3389/fmicb.2023.1156292. PMID: 37538841; PMCID: PMC10394836.
|
|
|
Fig. 3 In the legend, Fig 3c is missing. How were the N0/Nt numbers calculated? It is not possible to calculate reliable phage titers only from spot tests!
|
|
(c) is added .
Calculating the logarithmic decrease of phage titer compared to the initial titer is a robust way to quantify and interpret the impact of environmental stress. Logarithmic values transform significant differences in titer into manageable scales, making it easier to compare phage stability or resistance under different environmental conditions.. The primary concern is how much the titer decreases after stress exposure, not the absolute numbers. A focus on the logarithmic decrease ensures the emphasis remains on the effect of stress rather than variations in starting or ending titers. For example:
Was repeated with Double agar-overlay. |
|
|
Fig. 4 Same problems as for Fig. 3. In the legend, Fig 4c is missing. How were the N0/Nt numbers calculated? It is not possible to calculate reliable phage titers only from spot tests!
|
|
c) is added .
Calculating the logarithmic decrease of phage titer compared to the initial titer is a robust way to quantify and interpret the impact of environmental stress. Logarithmic values transform significant differences in titer into manageable scales, making it easier to compare phage stability or resistance under different environmental conditions.. The primary concern is how much the titer decreases after stress exposure, not the absolute numbers. A focus on the logarithmic decrease ensures the emphasis remains on the effect of stress rather than variations in starting or ending titers. For example:
Was repeated with Double agar-overlay. |
|
|
Fig. 5 It is not possible to calculate reliable phage titers only from spot tests! Apply plating assays |
|
Calculating the logarithmic decrease of phage titer compared to the initial titer is a robust way to quantify and interpret the impact of environmental stress. Logarithmic values transform significant differences in titer into manageable scales, making it easier to compare phage stability or resistance under different environmental conditions.. The primary concern is how much the titer decreases after stress exposure, not the absolute numbers. A focus on the logarithmic decrease ensures the emphasis remains on the effect of stress rather than variations in starting or ending titers. For example:
Was repeated with Double agar-overlay. |
|
|
Some spelling errors have to be corrected • P3: Transmission Electron Microscopy
|
Line 74 |
Corected |
Line 90 |
|
Quantitatively
|
Line 78 |
Corected |
Line 95 |
|
• P4 : Nucleotide Blast
|
Line 147 |
Corrected |
Line 107 |
|
Check the uniformity of the phage names: sometimes it is Ea PF7 and Ea PF9, sometimes EA Pf7 and EA PF9 resp. |
|
Checked and corrected |
Lines : 196-200, 211, 228-231, 244-247, 287 |
|
Check the word phage titer instead of phage titter |
|
Checked and corrected |
Lines:101-103, 144,256,262 |
|

Reviewer 3 Report
Comments and Suggestions for Authors
The research article by Lomadze et al. on "Characterization of newly isolated Erwinia amylovora Loessnervirus-like bacteriophages from Hungary" describes the potential of two Erwinia phages in controlling the bacterial species. The basic characterization of phages is studied well but the applicability of these phages to plant diseases is not studied as claimed by the authors.
1. Both the phages are morphologically and genomically similar. These phages' lytic activity (host range) differs, including plaque formation. What is the 2% mismatch or difference in their genomes (98% similarity)?
Comments:
1. Line no. 14: What is the number of strains?
2. Line no. 30: Name the diseases caused by Ea.
3. Line no. 37,38: What are the chemicals or products used to control? What type of resistance developed?
4. Line no. 44-46: It looks more general. Write about the advantages of using phages against plant diseases.
5. Line no. 49: Be consistent with writing Ea (italics or not).
6. Line no. 53-55: How many strains of Ea cause diseases in plants?
7. Line no. 71: Write the exact location. How many samples are collected?
8. Line no. 75: How were the phages prepared and what concentration?
9. Line no. 107: Why LB broth? It is usually done using a buffer.
10. Line no. 138: What is the length of the raw reads and coverage?
11. Line no. 149: Results should start with the phage isolation process. How many samples were processed, how many phages were isolated, and how were these two phages chosen for this study? What was the host strain used for enrichment?
12. Line no. 153: Table 2 is not necessary here. Add as supplemental.
13. Line no. 155: Is it both the phages? Need more clarity in writing.
14. Line no. 159: The plaque size looks the same in Figure 1 a,b.
15. Line no. 171: Is it a cocktail? If not, explain the activity of the phages individually.
16. Table 3: It should be self-explanatory. What are the numbers inside the table?
17. Figure 2 A, B: Definitely, there is an emergence of resistant mutants after 15 hours. Did the authors observe the same in overlay plates? Why not test the cocktail (mix two phages)?
18. Figure C, D: The X-axis should be identical to compare.
19. Figures 3 and 4: Why more than 60 min incubation is necessary and what difference is it making?
20. Line no. 223: Genome analysis needs comparative analysis of these genomes. Only a 2% difference was noted between the phage genomes, which causes substantial differences in their lytic activity. So the question is on the receptor-binding domain.
21. The discussion fails to address the important findings in the results. 1) How do both these phages differ, especially the genome, 2) Why phage-resistant mutants, 3) How and why these two phages can be used in plant diseases?
Author Response
Referee 2 |
|||
What is the number of strains
|
Line no. 14: |
The number of strains is added
|
Line 15 |
Name the diseases caused by Ea.
|
Line no. 30 |
Done |
Line 32 |
Both the phages are morphologically and genomically similar. These phages' lytic activity (host range) differs, including plaque formation. What is the 2% mismatch or difference in their genomes (98% similarity)?
|
|
Added more information in manucript |
Lines 304-315 |
What are the chemicals or products used to control? What type of resistance developed?
|
Line no. 37,38 |
Information is added in manuscript. |
Lines 38- 46 |
It looks more general. Write about the advantages of using phages against plant diseases.
|
Line no. 44-46 |
Information is added in manuscript |
51-57 |
Be consistent with writing Ea (italics or not). |
Line no. 49 |
Corrected |
Line 62 |
How many strains of Ea cause diseases in plants? |
Line no. 53-55: |
To provide a precise count of pathogenic strains, it would require an exhaustive review of all identified E. amylovora strains. However, nearly all naturally occurring strains studied are considered pathogenic to some degree, given the bacterium's reputation as the causative agent of fire blight. There are a few reports of strains with reduced or no pathogenicity, typically due to mutations or loss of critical virulence genes.
Piqué N, Miñana-Galbis D, Merino S, Tomás JM. Virulence Factors of Erwinia amylovora: A Review. Int J Mol Sci. 2015 Jun 5;16(6):12836-54. doi: 10.3390/ijms160612836. PMID: 26057748; PMCID: PMC4490474.
Wang D, Korban SS, Pusey PL, Zhao Y. AmyR is a novel negative regulator of amylovoran production in Erwinia amylovora. PLoS One. 2012;7(9):e45038. doi: 10.1371/journal.pone.0045038. Epub 2012 Sep 18. PMID: 23028751; PMCID: PMC3445560.
|
|
Write the exact location. How many samples are collected? |
Line no. 71 |
Done in manuscript Answer: Samples for bacteriophage isolation were collected in Hungary, Pecs, from the apple gardens, where fire blight symptoms were seen. 10 samples of apples and soil were collected. |
Lines 79-81 |
How were the phages prepared and what concentration |
Line no. 75 |
Information is addeed in manuscript |
Lines 91-94 |
Why LB broth? It is usually done using a buffer |
Line no. 107 |
Answer : While buffer solutions are commonly used, there are numerous instances where LB broth or other media are utilized, as documented in various studies (Erol HB et al., 2022; Zurabov F et al., 2021; Jamal M et al., 2015; Capra ML et al., 2006). Notably, in some cases, the type of suspension medium is not explicitly specified for temperature-related phage tests (Peng Q et al., 2023; Kim S.G et al., 2022). Given this precedent, using media broth should not pose any issues. Furthermore, commercially available phage preparations, such as those from the Eliava Institute, are often suspended in broth (Villarroel J et al., 2017), reinforcing the suitability of broth media for this purpose.
Erol HB, Kaskatepe B, Ozturk S, Safi Oz Z. The comparison of lytic activity of isolated phage and commercial Intesti bacteriophage on ESBL producer E. coli and determination of Ec_P6 phage efficacy with in vivo Galleria mellonella larvae model. Microb Pathog. 2022 Jun;167:105563. doi: 10.1016/j.micpath.2022.105563. Epub 2022 May 2. PMID: 35513294. Zurabov, F., Zhilenkov, E. Characterization of four virulent Klebsiella pneumoniae bacteriophages, and evaluation of their potential use in complex phage preparation. Virol J 18, 9 (2021). https://doi.org/10.1186/s12985-020-01485-w. Jamal M, Hussain T, Das CR, Andleeb S. Characterization of Siphoviridae phage Z and studying its efficacy against multidrug-resistant Klebsiella pneumoniae planktonic cells and biofilm. J Med Microbiol. 2015 Apr;64(Pt 4):454-462. doi: 10.1099/jmm.0.000040. Epub 2015 Feb 13. PMID: 25681321. Capra ML, Quiberoni A, Reinheimer J. Phages of Lactobacillus casei/paracasei: response to environmental factors and interaction with collection and commercial strains. J Appl Microbiol. 2006 Feb;100(2):334-42. doi: 10.1111/j.1365-2672.2005.02767.x. PMID: 16430510. Peng Q, Ma Z, Han Q, Xiang F, Wang L, Zhang Y, Zhao Y, Li J, Xian Y, Yuan Y. Characterization of bacteriophage vB_KleM_KB2 possessing high control ability to pathogenic Klebsiella pneumoniae. Sci Rep. 2023 Jun 17;13(1):9815. doi: 10.1038/s41598-023-37065-5. Kim, S.G.; Lee, S.B.; Jo, S.J.; Cho, K.; Park, J.K.; Kwon, J.; Giri, S.S.; Kim, S.W.; Kang, J.W.; Jung, W.J.; et al. Phage cocktail in combination with kasugamycin as a potential treatment for fire blight caused by Erwinia amylovora. Antibiotics 2022, 11, 1566. https://doi.org/10.3390/antibiotics11111566
Villarroel J, Larsen MV, Kilstrup M, Nielsen M. Metagenomic Analysis of Therapeutic PYO Phage Cocktails from 1997 to 2014. Viruses. 2017 Nov 3;9(11):328. doi: 10.3390/v9110328. PMID: 29099783; PMCID: PMC5707535.
|
|
What is the length of the raw reads and coverage?
|
. Line no. 138 |
Information is added in manuscript:
Raw reads Coverage |
Line 163 Line 295 |
Results should start with the phage isolation process. How many samples were processed, how many phages were isolated, and how were these two phages chosen for this study? What was the host strain used for enrichment.
|
Line no. 149: |
Information is added in manuscript |
Lines 169-176 |
Table 2 is not necessary here. Add as supplemental. |
Line no. 153 |
Table 2 is removed |
|
Is it both the phages? Need more clarity in writing.
|
Line no. 155 |
Table 2 is removed |
|
The plaque size looks the same in Figure 1 a,b.
|
Line no. 159 |
In case of Ea PF 9 plaques were more unform. Formulation is changed |
Line 180 |
Is it a cocktail? If not, explain the activity of the phages individually |
Line no. 171: |
Corrected |
Line 193 and Lines 215-218 |
Table 3: It should be self-explanatory. What are the numbers inside the table?
|
|
Done and renamed as Table 2
|
Lines 204-209 |
Figure 2 A, B: Definitely, there is an emergence of resistant mutants after 15 hours. Did the authors observe the same in overlay plates? Why not test the cocktail (mix two phages)? |
|
After 24 hours of incubation, no mutant bacterial growth was observed with either the double-agar overlay method or the spot test method, making it challenging to conclude at this stage. Usually, phage cocktails contain several phages. The next phase of the investigation will involve testing these two phages alongside existing phages, such as EaH1, EaH2, and L1, to explore the potential for cocktail formulation and assess any antagonistic or synergistic interactions. |
Lines 218-226 |
18. Figure C, D: The X-axis should be identical to compare.
|
|
Figure changed |
|
Figures 3 and 4: Why more than 60 min incubation is necessary and what difference is it making?
|
|
We conducted phage inoculations at various temperatures, extending the incubation period from 60 to 120 minutes. This adjustment was made to observe the temperature-dependent dynamics of phage behavior over a prolonged incubation period, potentially providing more insight into temperature effects on phage activity and host interactions. In the case of Ea PF9, the data revealed that after 90 minutes of incubation, the phage titer dropped to 0, indicating complete inactivation. This outcome aligns with the rapid loss of viability observed at higher temperatures, particularly at 55°C. . |
Lines 247-250 |
Genome analysis needs comparative analysis of these genomes. Only a 2% difference was noted between the phage genomes, which causes substantial differences in their lytic activity. So the question is on the receptor-binding domain. |
Line no. 223 |
Done comparative analysis of these genomes |
Lines 304-315 |
21. The discussion fails to address the important findings in the results. 1) How do both these phages differ, especially the genome, 2) Why phage-resistant mutants, 3) How and why these two phages can be used in plant diseases ?
|
|
1) Added in manuscript
2) Added in manuscript
3) Adde in manuscript in results and discusions and conclusion
|
Lines 304-315 Lines 218-325
|

Round 2
Reviewer 2 Report
Comments and Suggestions for Authors
Unfortunately, the authors did not add any new data (except one TEM picture of a contracted phage) or more clear experimental setups to the paper, the figs 2c and 2d are worse than before; the numbers on the abscissae are confusing. This is also the case for figs. 2 a and b.
The critical questions in the first version were only answered by citing other authors not by own data or discussions. The new references are not consistent and have to be checked (e.g ref 13 in line 42 has to be 18).
Do I get this right and there is no discussion section at all in the new version? The data HAVE to be discussed thoroughly.
Reviewer 3 Report
Comments and Suggestions for Authors
The authors have made substantial corrections. However, there are some concerns about the author's response;
1. Line no. 39: Differentiate antibiotics and chemicals separately.
2. Line no. 251-253: Resistant mutants develop after 14 hours, which means the authors should be able to isolate them from the microtiter plate. Isolate them and perform a spot test.
3. Line no. 348: This is not true. Phage EaPF9 also contains homing endonuclease, which is not annotated. i.e., ORF_2 (467-970) is a homing endonuclease with 51% identity to EaPF7.
4. Genome map is poorly presented. Where is comparative genomics?
